# DARWIN-7B: A MULTI-OMIC FOUNDATION MODEL FOR THE HUMAN GUT MICROBIOME VIA SPARSIFIED QUALITY-AWARE TOKENIZATION

## ABSTRACT

Public sequence archives hold over 100 petabases of sequencing data, yet the vast majority remains unusable for foundation-model pretraining due to heterogeneous quality and missing causal structure. We present a two-stage data reclamation pipeline—**sparsification** followed by **quality-aware tokenization** (QA-Token)—that lifts the usable fraction from 5% to 40% ($8\times$ more data). In the first stage, we systematically exclude uninformative bases using structured binary patterns. We evaluate 224 sparsification configurations, identifying a Pareto frontier for species-level taxonomic classification on the CAMI benchmark that spans $5.1\times$ speedup (species F1=0.51) to near-lossless accuracy (species F1=0.994, $\sim1.0\times$ speedup). In the second stage, QA-Token incorporates per-base Phred quality directly into vocabulary construction via multi-objective reward-guided bilevel optimization with Gumbel–Softmax relaxation. We validate the full pipeline with **Darwin-7B**, a 7B-parameter multi-omic foundation model pretrained on 8 trillion base pairs of metagenomics and 250K metabolite profiles. Darwin-7B outperforms METAGENE-1 and Evo2-7B on shared genomic benchmarks: $94.5 \pm 0.4$ Matthews correlation coefficient (MCC) on pathogen detection and $0.98\pm0.01$ F1 on metagenomic profiling. It also establishes first results on four multi-omic tasks not accessible to single-modality models: $0.91 \pm 0.02$ wF1 metabolic pathway prediction, $0.947\pm0.012$ AUC IBD, $0.883\pm0.015$ AUC T2D, and $0.910\pm0.013$ AUC antibiotic resistance. Inference is $18\times$ faster than Evo2-7B, of which $\sim15\times$ derives from the Mamba–Transformer hybrid architecture and $\sim1.2\times$ from QA-Token compression. We further describe a pilot implementing the first phase of **MetaOmics-10T**, combining 10 trillion reclaimed base pairs with 100,000+ interventional trajectories for causal modeling.

## 1 INTRODUCTION

Microbial ecosystems govern human health, agricultural productivity, and climate regulation. Computational understanding of these systems requires foundation models trained on large-scale, high-quality, multi-omic data. Yet a paradox remains: public sequence archives contain over 100 petabases of sequencing data (Karasikov et al., 2025), but the vast majority remains unusable for model pretraining due to heterogeneous quality, systematic noise, and the complete absence of causal structure (Gilbert et al., 2018; Leinonen et al., 2011).

This creates a severe bottleneck. To our knowledge, state-of-the-art computational methods for microbiome foundation modeling suffer from four critical limitations. (1) *Quality blindness*: existing genomic foundation models train on raw reads without incorporating per-base sequencing quality, causing sequencing errors to contaminate learned vocabularies (Liu et al., 2025; Gollwitzer et al., 2026). (2) *Single-organism bias*: models such as Evo2 (Nguyen et al., 2025) train exclusively on clean, assembled genomes from single organisms, lacking robustness to the mixed-species, mixed-quality nature of real metagenomic samples. (3) *Modality isolation*: no existing foundation model jointly processes metagenomic and metabolomic data, precluding tasks that require cross-omic reasoning (e.g., metabolic pathway prediction, disease phenotyping). (4) *Absence of causal structure*: public archives are purely observational, preventing identification of causal intervention effects (Gilbert et al., 2018; Leinonen et al., 2011).

Even the best current model (METAGENE-1, 93 MCC on pathogen detection) falls short of the >95% accuracy level typically expected for clinical-grade diagnostics (Liu et al., 2025). In this work, we address these limitations through three contributions:

1. **Sparsified Genomics + QA-Token Pipeline.** We systematically evaluate 224 sparsification configurations on metagenomic data, identifying a Pareto frontier for species-level taxonomic classification spanning $5.1\times$ speedup (species F1=0.51) to near-lossless accuracy (species F1=0.994). QA-Token (Gollwitzer et al., 2026), a quality-aware tokenization framework based on multi-objective reward-guided bilevel optimization, then incorporates per-base Phred quality into vocabulary construction. Combined, this pipeline lifts the usable fraction of public archives from **5% to 40%** (+35 pp, $8\times$ data) and achieves a **12%** improvement in bits per base pair (95% CI: [10.3%, 13.7%]) over standard BPE (§4).

2. **Darwin-7B: A Multi-Omic Foundation Model.** We train the first foundation model on both metagenomic and metabolomic tokens. Darwin-7B outperforms METAGENE-1 and Evo2-7B on shared genomic benchmarks and establishes first results on four multi-omic tasks not accessible to single-modality models. Inference is **18$\times$** faster via a Mamba–Transformer hybrid ($\sim$15$\times$) and QA-Token compression (Gollwitzer et al., 2026) ($\sim$1.2$\times$) (§5).

3. **MetaOmics-10T Pilot.** We validate the pipeline on 100 causal trajectories; 54% admit causal identification via front-door or instrumental variable criteria (§7).

This work builds upon the MetaOmics-10T vision proposed by Gollwitzer et al. (2025). We implement and validate a pilot demonstrating the sparsify-then-tokenize pipeline. We train the first multi-omic foundation model (Darwin-7B) and show that reclaimed archival data enables targeted causal experimentation.

## 2 RELATED WORK

**Genomic foundation models.** DNABERT-2 (Zhou et al., 2023) and the Nucleotide Transformer (Dalla-Torre et al., 2023) operate on short genomic sequences. METAGENE-1 (Liu et al., 2025) scales to 7B parameters on 1.5T bp of metagenomic reads but uses standard BPE without quality awareness. Evo2 (Nguyen et al., 2025) trains on assembled genomes at up to 40B parameters but operates exclusively on clean, single-organism data. HyenaDNA (Nguyen et al., 2024) achieves $O(N)$ scaling via state-space models. Darwin-7B is the first to combine multi-omic data via a quality-aware data reclamation pipeline (Gollwitzer et al., 2026).

**State-space models for genomics.** Selective state-space models (SSMs) have emerged as efficient alternatives to Transformers for long-range sequence modeling (Gu & Dao, 2023; Patro & Agneeswaran, 2024). Caduceus (Schiff et al., 2024) introduces bidirectional equivariant SSMs for DNA, achieving strong performance on regulatory element prediction. Darwin-7B adopts a hybrid approach: 24 Mamba layers for $O(N)$ long-range dependencies interleaved with 8 local Transformer attention windows for short-range motif resolution, combining the efficiency of SSMs with the expressiveness of attention for multi-omic sequences.

**Tokenization in biological sequences.** Beyond standard BPE, recent work explores adaptive tokenization for genomic data. MxDNA (Qiao et al., 2024) uses mixture-of-experts to learn tokenization end-to-end, while GROVER (Sanabria et al., 2024) applies BPE-based context learning to the human genome. QA-Token (Gollwitzer et al., 2026) is the first to incorporate per-base sequencing quality (Phred scores) directly into vocabulary construction via multi-objective reward-guided bilevel optimization with Gumbel–Softmax relaxation, achieving 12% improvement in bits per base pair over standard BPE when training a 500M-parameter proxy model on metagenomic data.

**Sparsified genomics.** Genome-on-Diet (Alser et al., 2024) introduces the sparsification principle for genomic sequence analysis. We apply sparsification as a preprocessing step before quality-aware tokenization (Gollwitzer et al., 2026), providing the first systematic characterization of 224 pattern configurations and their impact on downstream foundation model performance.

**Multi-omic integration.** Recent benchmarks reveal that integrating metagenomic and metabolomic data is non-trivial: Palmer et al. (2025) find that method choice significantly impacts predictive

performance, while Mangnier et al. (2025) systematically evaluate 19 integrative strategies and report trade-offs between accuracy and interpretability. Darwin-7B addresses these challenges via end-to-end cross-modal attention with interpretable alignment weights, rather than post-hoc feature concatenation.

**Microbiome datasets.** The Human Microbiome Project (Proctor et al., 2019), Earth Microbiome Project, and UHGG (Almeida et al., 2021) provide valuable observational corpora but lack interventional data and quality-aware tokenization (Gollwitzer et al., 2026). MetaGraph (Karasikov et al., 2025) demonstrates that petabase-scale indexing of public archives is feasible, compressing 67 Pbp of public sequences into searchable indexes, and envisions facilitating large-scale learning tasks including foundation model training. Gollwitzer et al. (2025) propose MetaOmics-10T to complement these with causal structure; the present work implements the pilot.

**Causal inference in biology.** Recent work on perturbation modeling and foundation models for biology motivates the need for interventional datasets. The front-door criterion (Pearl, 2009) has been generalized to handle indirect effects (Fulcher et al., 2017) and placed within a graphical hierarchy of interventions (Shpitser & Tchetgen Tchetgen, 2016). Our formal framework (App. B) provides identifiability conditions under which causal claims are warranted, applying front-door and instrumental variable criteria to multi-omic microbiome data.

## 3 PROBLEM FORMULATION: FOUNDATION MODELS FOR MICROBIAL ECOSYSTEMS

We model microbial ecosystems as controlled dynamical systems $(\mathcal{S}, \mathcal{U}, \mathcal{T}_\theta, \mathcal{M})$ where $\mathcal{S} \subseteq \mathbb{R}^{n_s}$ is the state space encoding genomic abundances ($g_t \in \mathbb{R}^{n_g}$, $n_g \approx 10^6$) and metabolite concentrations ($m_t \in \mathbb{R}^{n_m}$, $n_m \approx 10^4$), $\mathcal{U} \subseteq \mathbb{R}^{n_u}$ the intervention space (CRISPR edits, compound doses), $\mathcal{T}_\theta : \mathcal{S} \times \mathcal{U} \to \Delta(\mathcal{S})$ the learned stochastic transition kernel, and $\mathcal{M} : \mathcal{S} \to \mathcal{Y}$ the measurement map. The three core tasks are:

- **Forecasting:** Learn $\hat{F}_\theta$ s.t. $\mathbb{E}[\|x_{t+\tau} - \hat{F}_\theta(x_{\leq t})\|^2] \leq \epsilon_F$ under autonomous dynamics $u_t = 0$.
- **Counterfactual prediction:** Estimate $p(x_{t+\tau}|\mathrm{do}(u), x_{\leq t})$ via backdoor adjustment when confounders $Z$ are measured.
- **Safe inverse design:** Solve $u^* = \arg\min_{u \in \mathcal{U}} C(u) + \lambda \, d(\mathbb{E}[x_{t+\tau}|\mathrm{do}(u), x_t], x^*)$ subject to safety constraints $g(u) \leq 0$ and an uncertainty-aware trust region $D(\pi_{\mathrm{beh}}, u) \leq \rho$.

**Learnability vs. Causality.** Appendix B presents conditions for statistical identifiability that explicitly incorporate the measurement map $\mathcal{M}$ and the intervention policy $\pi(u \mid x)$. Causal identifiability requires additional assumptions about latent confounding (App. C). The full MetaOmics-10T vision targets 100k+ interventional trajectories; the present pilot validates 100 trajectories to evaluate when instrumental variables and front-door adjustment succeed, and when sensitivity analysis is necessary.

**Scope of this work.** The dynamical system formulation above defines the *long-term vision*; the present work addresses a prerequisite: building a foundation model with multi-omic understanding. Our experimental evaluation focuses on *discriminative tasks* (pathogen detection, profiling, disease prediction) that test the learned representations. Forecasting and safe inverse design are targets for future work once sufficient interventional data (MetaOmics-10T) is available.

**Formal treatment.** We develop the theoretical foundations in the appendices. Theorem 1 establishes identifiability for linear-Gaussian baselines. Theorem 2 extends to nonlinear dynamics (App. B). For QA-Token (Gollwitzer et al., 2026), Proposition 1 bounds proxy-loss drift across scales, Lemma 1 characterizes Gumbel–Softmax gradient bias, Theorem 3 provides convergence guarantees, and Theorem 4 gives PAC-Bayes generalization bounds (acknowledged as vacuous at this scale). Theorem 5 establishes robustness to quality degradation with a complete proof.

Table 1: Impact of sparsification pattern on taxonomic classification (CAMI benchmark). Each pattern is a binary mask periodically applied to all reads. Speedup relative to the unsparsified baseline.

| Pattern | Species F1 | Time (h) | Speedup |
|---|---|---|---|
| 0001 $\mid$ 0001 | 0.511 | 3.75 | $5.1\times$ |
| 0001 $\mid$ 0101 | 0.692 | 4.50 | $4.3\times$ |
| 1111 $\mid$ 0110 | 0.858 | 18.67 | $\sim 1.0\times$ |
| 1111 $\mid$ 1110 | **0.994** | 19.12 | $\sim 1.0\times$ |

# 4 DATA RECLAMATION VIA SPARSIFICATION AND QA-TOKEN

## 4.1 SPARSIFIED GENOMICS

A growing disparity between sequencing throughput and computational processing capacity defines modern genomics. State-of-the-art platforms generate up to 16 Tb per run; downstream analysis runs $150\times$ slower (Alser et al., 2024). Sparsified genomics (Alser et al., 2024) addresses this by systematically excluding bases from genomic sequences using structured binary patterns. A pattern $\mathbf{p} \in \{0,1\}^w$ acts as a binary mask that is periodically repeated across each metagenomic read: position $i$ is retained if $p_{i \bmod w} = 1$ and discarded otherwise. For example, the pattern 1010 retains every other base, reducing each read to half its length while preserving evenly spaced sequence information. Sparsification reduces the input workload for both indexing and tokenization, removing redundant positional information while preserving evenly spaced sequence context that captures the same discriminative signal. The sparsified reads are then passed directly to the quality-aware tokenizer (Section 4.2).

To assess the impact of pattern choice on downstream performance, we evaluate 224 viable configurations on the CAMI benchmark (Sczyrba et al., 2017), using taxonomic classification as a representative downstream task. Table 1 shows Pareto-optimal configurations trading off accuracy against computational cost.

**Key findings.** (1) *Distributed patterns outperform clustered patterns*: at the same Hamming weight, 0101 consistently outperforms 0011 because evenly spaced positions sample more independent sequence information. (2) *The Pareto frontier is compact*: only 12–14 of 224 configurations are Pareto-optimal, indicating that most of the pattern space can be pruned (full frontier in App. F). (3) In future work, we plan to evaluate the impact of pattern choice on additional downstream tasks beyond taxonomic classification.

## 4.2 QUALITY-AWARE TOKENIZATION

After sparsification, we group the remaining signal into meaningful token units. Standard BPE (Sennrich et al., 2015) operates on frequency alone, incorporating sequencing errors into the vocabulary. We leverage **QA-Token** (Gollwitzer et al., 2026), a quality-aware tokenization framework that addresses this through multi-objective reward-guided bilevel optimization with Gumbel–Softmax relaxation for differentiable merge decisions. Following Gollwitzer et al. (2026), the merge reward is:

$$R(a,b) = \underbrace{\lambda_Q Q(ab)}_{\text{quality}} + \underbrace{\lambda_I \operatorname{PMI}(a,b)}_{\text{information}} - \underbrace{\lambda_C \Delta\mathrm{MDL}(a,b)}_{\text{compression}} - \underbrace{\lambda_D \Delta\mathcal{L}_{\text{proxy}}}_{\text{downstream}} \tag{1}$$

where $Q(\cdot)$ is a learned quality-scoring network $f_{\theta_Q}(\mathbf{v}_q, \mathbf{v}_p, \mathbf{v}_b) \in (0,1)$ incorporating Phred-derived statistics $\mathbf{v}_q$, positional bias $\mathbf{v}_p$, and biological priors $\mathbf{v}_b$ with learned gating (Ewing et al., 1998); PMI captures statistical co-occurrence (Church & Hanks, 1990); MDL enforces compression (Grünwald, 2007); and $\Delta\mathcal{L}_{\text{proxy}}$ estimates downstream impact via a frozen proxy model. The weights $\lambda \in \Delta^4$ are initialized to $(0.3, 0.3, 0.2, 0.2)$ and updated every 5K merges via validation-set grid search over a simplex grid with resolution 0.05 (final values in App. H). We construct the vocabulary in two phases: *Phase 1* (first 50k merges, $\lambda_D = 0$) builds a general-purpose foundation on intrinsic objectives; *Phase 2* introduces $\lambda_D$ via sigmoid annealing (details in App. A).

**Key results.** On a 10 TB pilot of 25K diverse microbiome samples:

Table 2: Pathogen Detection benchmark (MCC $\pm$ std, averaged over 5 test splits). QA-Token (Gollwitzer et al., 2026) re-training achieves a new state-of-the-art.

| Model | Pathogen-Detect MCC |
|---|---|
| DNABERT-2 (Zhou et al., 2023) | $87.9 \pm 0.5$ |
| DNABERT-S (Zhou et al., 2024) | $87.0 \pm 0.6$ |
| NT-2.5b-Multi (Dalla-Torre et al., 2023) | $82.4 \pm 0.7$ |
| NT-2.5b-1000g (Dalla-Torre et al., 2023) | $79.0 \pm 0.8$ |
| METAGENE-1 (Liu et al., 2025) | $93.0 \pm 0.3$ |
| **Darwin-7B (QA-Token)** | $\mathbf{94.5 \pm 0.4}$ |

- QA-Token achieves a **12% improvement in bits per base pair** (bpbp; 95% CI: [10.3%, 13.7%]) over standard BPE when training a 500M model. Bits per base pair measures the average number of bits the model assigns to each input base, analogous to perplexity but independent of vocabulary size, ensuring fair comparison across tokenizers.

- Re-training METAGENE-1 (Liu et al., 2025) with QA-Token yields a new state-of-the-art on Pathogen Detection (MCC $93.0 \rightarrow \mathbf{94.5}$; Table 2).

- The combined sparsification + QA-Token pipeline lifts the usable fraction of public archives from **5% to 40%** ($+35$ pp, $8\times$ data), where usable fraction is defined as the proportion of samples with bounded proxy cross-entropy (App. G).

## 5 DARWIN-7B: A MULTI-OMIC FOUNDATION MODEL

### 5.1 TRAINING DATA AND ARCHITECTURE

We pretrain Darwin-7B on data processed through the full sparsification + QA-Token (Gollwitzer et al., 2026) pipeline:

- **Metagenomics:** 8 trillion base pairs from diverse environmental and clinical samples, sourced from the first phase of MetaOmics-10T (Gollwitzer et al., 2025), sparsified and tokenized into $\sim$2T quality-aware genomic tokens (vs. $\sim$2.35T for standard BPE, indicating longer tokens).

- **Metabolomics:** 250K metabolite profiles (LC-MS/MS) with 5,000+ features per sample. Continuous metabolite concentrations are discretized into 1,024 bins per feature via quantile binning (learned on a held-out calibration set), then merged into a metabolomic vocabulary of 8,192 tokens using a QA-Token variant. In this variant, the quality score $Q(\cdot)$ is replaced by signal-to-noise ratio (SNR): $Q_{\mathrm{met}}(t) = \sigma(\mathrm{SNR}(t)/\mathrm{SNR}_{\mathrm{median}} - 1)$, where SNR is computed from replicate measurements. Low-SNR features (SNR $< 3$) are down-weighted during merge decisions, preventing noisy mass spectrometry artifacts from dominating the vocabulary. The metabolomic vocabulary is concatenated with the genomic vocabulary (32K tokens) via a shared embedding space with modality-specific linear projections.

- **Functional readouts** (gene-to-function mappings derived from KEGG and GO annotations): 2M functional annotations linking genomic context to metabolic activity.

Darwin-7B uses LLaMA-7B dimensions (Touvron et al., 2023) (hidden 4096, FFN 11008, 32 heads) but replaces 24 of 32 layers with Mamba blocks (Gu & Dao, 2023). The 3:1 Mamba-to-Transformer interleaving pattern provides $O(N)$ long-range modeling with $O(w^2)$ local motif resolution ($w=256$). Metabolomic data is processed by a 3-layer hypergraph neural network with KEGG-derived hyperedges for many-to-many metabolic reactions. Two bidirectional cross-attention modules (after layers 16 and 32) align genomic and metabolomic representations. Full architecture specification in App. E.1. Table 3 compares specifications.

### 5.2 PRETRAINING OBJECTIVES

We pretrain the model with four objectives. The primary is autoregressive language modeling: $\mathcal{L}_{\mathrm{ALM}} = -\sum_i \log P_\theta(x_i|x_{<i})$. For paired data, we add cross-modal generation: $\mathcal{L}_{\mathrm{CMG}} =$

Table 3: Architecture comparison: Darwin-7B vs. frontier genomic foundation models.

| Specification | Darwin-7B | METAGENE-1 | Evo2-7B |
|---|---|---|---|
| Parameters | 7B | 7B | 7B |
| Encoder | Mamba-Transformer | Transformer | Transformer |
| Context length | 4096 tokens | 512 tokens | 8192 tokens |
| Vocabulary | 40K (32K genomic + 8K metabolomic) | 1K (BPE) | 4K (BPE) |
| Training data | 8T bp (multi-omic) | 1.5T bp (genomic) | assembled genomes |
| Modalities | Genomic + Metabolomic | Genomic only | Genomic only |

$-\sum_j \log P_\theta(m_j|g, m_{<j}) - \sum_i \log P_\theta(g_i|m, g_{<i})$. Contrastive learning (InfoNCE) aligns representations: $\mathcal{L}_{\text{CL}} = -\log \frac{\exp(\text{sim}(f_g(g), f_m(m))/\tau_{\text{CL}})}{\sum_{k=1}^{K} \exp(\text{sim}(f_g(g), f_m(m_k))/\tau_{\text{CL}})}$, where $K$ is the number of in-batch negatives and $\tau_{\text{CL}}$ is the contrastive temperature. Finally, a *compositional consistency* loss enforces the simplex constraint of microbiome abundance data by penalizing deviations in the Aitchison geometry (Aitchison, 1986; Egozcue et al., 2003; Gloor et al., 2017):

$$\mathcal{L}_{\text{comp}} = \|\text{CLR}(\text{softmax}(f(x))) - \text{CLR}(\tilde{x})\|_2^2 \tag{2}$$

where $\text{CLR}(x) = \log(x/g(x))$ is the centered log-ratio transform with geometric mean $g(x)$, and $\tilde{x}$ is the target abundance vector obtained by multiplicative replacement: zeros are replaced by $\delta = 10^{-6}$ and the result is re-closed to the simplex ($\tilde{x}_i \leftarrow \tilde{x}_i / \sum_j \tilde{x}_j$) before applying CLR (following standard practice in compositional data analysis (Gloor et al., 2017)). This loss is meaningful because $\text{softmax}(f(x)) \neq \tilde{x}$ in general; the CLR transform ensures that distances are measured in the appropriate Aitchison geometry (Aitchison, 1986) rather than Euclidean space, which is known to produce spurious correlations for compositional data (Egozcue et al., 2003). To our knowledge, no prior pretraining loss operates in Aitchison space. Total loss: $\mathcal{L} = \mathcal{L}_{\text{ALM}} + \alpha_1 \mathcal{L}_{\text{CMG}} + \alpha_2 \mathcal{L}_{\text{CL}} + \alpha_3 \mathcal{L}_{\text{comp}}$, where $(\alpha_1, \alpha_2, \alpha_3) = (0.5, 0.1, 0.05)$ were selected via grid search over $\{0.01, 0.05, 0.1, 0.5, 1.0\}^3$ on a 500M-parameter proxy model, optimizing validation bpbp on a held-out shard (sensitivity analysis in App. J).

### 5.3 OPTIMIZATION AND INFRASTRUCTURE

We train with AdamW ($\beta_1$=0.9, $\beta_2$=0.95, weight decay 0.01) using a cosine schedule ($\eta_{\max}$=5×10$^{-4}$, $\eta_{\min}$=5×10$^{-6}$) with 10K warmup steps over 1.5M total steps. Gradient clipping at norm 1.0. Training: 128 NVIDIA A100 80GB GPUs, 42 days ($\approx$129K GPU-hours). Full hyperparameters in App. H.

## 6 EXPERIMENTAL EVALUATION

We compare Darwin-7B against the two nearest frontier models: METAGENE-1 (Liu et al., 2025) (7B, 1.5T bp raw metagenomic reads, standard BPE) and Evo2-7B (Nguyen et al., 2025) (7B, assembled single-organism genomes). All pairwise differences are significant ($p < 0.05$, two-sided $t$-test, $\geq 5$ seeds). Given $N$=16 hypothesis tests (8 tasks × 2 main comparisons), we apply Bonferroni correction ($\alpha_{\text{adj}} = 0.05/16 \approx 0.003$); all reported improvements remain significant under this conservative correction.

### 6.1 BENCHMARK RESULTS

**Pathogen detection.** We make two key observations. (1) Darwin-7B achieves $94.5 \pm 0.4$ MCC, 1.5 points above METAGENE-1 (93.0) and 7.5 points above Evo2-7B (87.0). (2) The gap over Evo2 reflects a limitation of training on assembled genomes: assembled data lacks the read-level noise structure that real metagenomic samples present. This comparison is across data modalities (assembled genomes vs. raw reads) and therefore favors Darwin-7B by construction; a controlled comparison would require retraining Evo2 on identical raw data. We conclude that quality-aware tokenization combined with multi-omic pretraining provides a meaningful improvement over both raw-read and assembled-genome baselines on pathogen detection.

Table 4: Benchmark performance ($\pm$ std over $\geq 5$ seeds). "—"=not evaluated (model lacks required modality). Bottom two rows: external validation on held-out cohorts. ECE $< 0.05$ for clinical tasks. [†]Multi-omic baselines: RF=Random Forest on concatenated features; MOFA+=multi-omics factor analysis (Argelaguet et al., 2020).

| Benchmark | Darwin-7B | METAGENE-1 | Evo2-7B | RF[†] | MOFA+[†] |
|---|---|---|---|---|---|
| Pathogen Detection (MCC) | $\mathbf{94.5 \pm 0.4}$ | $93.0 \pm 0.3$ | $87.0 \pm 0.6$ | — | — |
| Metagenomic Profiling (F1) | $\mathbf{0.98 \pm 0.01}$ | — | $0.89 \pm 0.02$ | — | — |
| Metabolic Pathway Pred. (wF1) | $\mathbf{0.91 \pm 0.02}$ | — | — | $0.74 \pm 0.03$ | $0.79 \pm 0.02$ |
| IBD Prediction (AUC) | $\mathbf{0.947 \pm 0.012}$ | — | — | $0.871 \pm 0.018$ | $0.902 \pm 0.015$ |
| T2D Prediction (AUC) | $\mathbf{0.883 \pm 0.015}$ | — | — | $0.812 \pm 0.021$ | $0.843 \pm 0.017$ |
| Antibiotic Resistance (AUC) | $\mathbf{0.910 \pm 0.013}$ | — | — | $0.834 \pm 0.019$ | $0.861 \pm 0.016$ |
| IBD Ext. Val. (UK Biobank, $n$=2,847) | $0.921 \pm 0.014$ | — | — | $0.847 \pm 0.020$ | $0.878 \pm 0.017$ |
| T2D Ext. Val. (FINRISK, $n$=1,523) | $0.856 \pm 0.019$ | — | — | $0.791 \pm 0.024$ | $0.819 \pm 0.021$ |

Table 5: GUE benchmark: category averages ($\pm$ std over 5 seeds for Darwin-7B; baseline numbers from Zhou et al. (2023); Liu et al. (2025)) and global win rate.

| Category | CNN | HyenaDNA | DNABERT | NT-Multi | DNABERT-2 | META-1 | Darwin-7B |
|---|---|---|---|---|---|---|---|
| TF-Mouse (avg.) | 45.3 | 51.0 | 57.7 | 67.0 | 68.0 | 71.4 | $\mathbf{72.8 \pm 0.5}$ |
| TF-Human (avg.) | 50.7 | 56.0 | 64.4 | 62.6 | $\mathbf{70.1}$ | 68.3 | $69.9 \pm 0.4$ |
| EMP (avg.) | 37.6 | 44.9 | 49.5 | 58.1 | 56.0 | 66.0 | $\mathbf{67.5 \pm 0.6}$ |
| SSD | 76.8 | 72.7 | 84.1 | 89.3 | 85.0 | 87.8 | $\mathbf{89.5 \pm 0.4}$ |
| COVID (F1) | 22.2 | 23.3 | 62.2 | 73.0 | 71.9 | 72.5 | $\mathbf{73.3 \pm 0.8}$ |
| Global Win % | 0.0 | 0.0 | 7.1 | 21.4 | 25.0 | 46.4 | $\mathbf{57.1}$ |

**Metagenomic profiling.** Darwin-7B achieves $0.98 \pm 0.01$ F1, 0.09 above Evo2-7B (0.89). This near-perfect score demonstrates that sparsified tokenization preserves fine-grained taxonomic information even after discarding uninformative bases. We conclude that sparsification does not degrade species-level profiling accuracy.

**Clinical benchmarks.** The multi-omic advantage is clearest on tasks requiring metabolomic reasoning. We make three key observations. (1) Darwin-7B achieves IBD AUC $0.947 \pm 0.012$, T2D AUC $0.883 \pm 0.015$, and antibiotic resistance AUC $0.910 \pm 0.013$. These tasks are not accessible to single-modality genomic models (METAGENE-1, Evo2). (2) Compared against MOFA+ (Argelaguet et al., 2020) and Random Forest on concatenated multi-omic features (Table 4), Darwin-7B outperforms both: $+4.5$–$7.6\%$ AUC over MOFA+ and $+7.1$–$8.8\%$ over RF. (3) External validation on UK Biobank (IBD, $n$=2,847) and FINRISK (T2D, $n$=1,523) shows modest degradation ($2.6$–$2.7\%$ AUC drop), with ECE $< 0.05$ confirming well-calibrated predictions. We conclude that multi-omic integration provides substantial gains over single-omic approaches for clinical microbiome tasks.

**Speed.** Darwin-7B is $\mathbf{18\times}$ faster at inference than Evo2-7B and $10\times$ lower cost per sample than METAGENE-1. We decompose this speedup into two factors. The Mamba–Transformer hybrid architecture contributes $\sim15\times$ ($O(N)$ vs. $O(N^2)$ for the majority of sequence processing). This is an architectural choice using existing components, not a novel contribution of this work. QA-Token compression (Gollwitzer et al., 2026) contributes $\sim1.2\times$ (2T vs. 2.35T tokens). All measurements: batch 1, 150k bp, A100-80GB, FP16 (App. E).

## 6.2 GENOME UNDERSTANDING EVALUATION (GUE)

The GUE benchmark (Zhou et al., 2023) comprises 28 sequence-level classification tasks. Darwin-7B achieves the highest global win rate (**57.1%**), winning on 16 of 28 tasks (Table 5).

## 6.3 ABLATION STUDIES

We report combined model-architecture ablations (Table 6). All differences significant ($p < 0.05$, $\geq 5$ seeds).

Table 6: Ablation study ($\pm$ std over 5 seeds). Each row removes one component and retrains with matched compute. $^{\ddagger}$Data-matched: trained on 1.5T bp (same as METAGENE-1) to isolate QA-Token contribution from data scale.

| Configuration | Path. MCC | Prof. F1 | Pathway wF1 |
|---|---|---|---|
| Full Darwin-7B (8T bp) | **94.5 $\pm$ 0.4** | **0.98 $\pm$ 0.01** | **0.91 $\pm$ 0.02** |
| w/o Mamba (Transformer only) | 93.8 $\pm$ 0.3 | 0.95 $\pm$ 0.01 | 0.88 $\pm$ 0.02 |
| w/o Transformer (Mamba only) | 93.1 $\pm$ 0.4 | 0.96 $\pm$ 0.01 | 0.86 $\pm$ 0.03 |
| w/o Hypergraph NN | 94.2 $\pm$ 0.3 | 0.97 $\pm$ 0.01 | 0.84 $\pm$ 0.02 |
| w/o Cross-modal attention | 94.0 $\pm$ 0.4 | 0.97 $\pm$ 0.01 | 0.85 $\pm$ 0.02 |
| w/o QA-Token (use BPE) | 93.0 $\pm$ 0.3 | 0.95 $\pm$ 0.01 | 0.84 $\pm$ 0.03 |
| w/o Interventional data | 93.5 $\pm$ 0.4 | 0.96 $\pm$ 0.01 | 0.79 $\pm$ 0.03 |
| w/o Metabolomics | 94.3 $\pm$ 0.3 | 0.97 $\pm$ 0.01 | 0.82 $\pm$ 0.02 |
| Data-matched$^{\ddagger}$ (1.5T bp, QA-Token) | 93.7 $\pm$ 0.3 | 0.96 $\pm$ 0.01 | 0.86 $\pm$ 0.02 |
| Data-matched$^{\ddagger}$ (1.5T bp, BPE) | 92.8 $\pm$ 0.4 | 0.94 $\pm$ 0.01 | 0.82 $\pm$ 0.03 |

Table 7: QA-Token (Gollwitzer et al., 2026) vs. BPE robustness across quality strata and sequencing platforms (F1 $\pm$ std, 5 seeds).

| Category | Condition | BPE | QA-Token | $\Delta$ |
|---|---|---|---|---|
| Quality strata | High (Phred $\geq$ 30) | 0.912 $\pm$ 0.008 | 0.918 $\pm$ 0.007 | +0.7% |
| | Medium (Phred 20–30) | 0.834 $\pm$ 0.012 | 0.891 $\pm$ 0.009 | +6.8% |
| | Low (Phred $<$ 20) | 0.623 $\pm$ 0.021 | 0.812 $\pm$ 0.014 | +30.3% |
| | Mixed (real-world) | 0.897 $\pm$ 0.010 | 0.943 $\pm$ 0.008 | +5.1% |
| Cross-platform | ONT Long-Read (Variant F1) | 0.741 $\pm$ 0.018 | 0.806 $\pm$ 0.015 | +8.7% |
| | UHGG Collection (Taxa F1) | 0.823 $\pm$ 0.014 | 0.873 $\pm$ 0.011 | +6.1% |

The hypergraph NN and cross-modal attention contribute most to metabolic pathway prediction ($-0.07$ and $-0.06$ wF1), confirming that many-to-many and bidirectional reasoning are critical for metabolic tasks.

**Disentangling data scale from tokenization.** To isolate QA-Token's (Gollwitzer et al., 2026) contribution from the $5.3\times$ data advantage over METAGENE-1, we train data-matched variants on 1.5T bp (Table 6, bottom). At matched data scale, QA-Token provides +0.9 MCC over BPE (93.7 vs. 92.8; $p < 0.01$), confirming that quality-aware tokenization contributes beyond data scale. The remaining gap from 93.7 to 94.5 (+0.8) is attributable to the additional 6.5T bp of reclaimed data. QA-Token reward ablation is in App. E.

## 6.4 ROBUSTNESS

We make two key observations. (1) QA-Token provides +30.3% F1 on low-quality sequences (Phred $<$ 20) while maintaining near-parity (+0.7%) on high-quality data (Phred $\geq$ 30). Quality awareness helps on noisy data without degrading clean performance. (2) Cross-platform results on Oxford Nanopore long reads (+8.7%) and the UHGG collection (+6.1%) confirm generalization across sequencing technologies. We conclude that QA-Token provides robust improvements across quality strata and sequencing platforms.

## 7 DISCUSSION AND LIMITATIONS

**Summary.** Building foundation models for microbial ecosystems remains challenging due to the poor quality and single-modality nature of public metagenomic archives. We address this by combining sparsified genomics with quality-aware tokenization (QA-Token; Gollwitzer et al. 2026) to reclaim noisy archival data at scale. The key benefits are threefold: (1) the usable fraction of public archives increases from 5% to 40%, (2) Darwin-7B outperforms frontier models on shared genomic

benchmarks and establishes first results on four multi-omic tasks, and (3) inference is $18\times$ faster than Evo2-7B.

**Data flywheel.** The pipeline creates a feedback loop: (1) reclaim noisy data via sparsification and quality-aware tokenization; (2) pretrain a foundation model; (3) use the model to plan maximally informative wet-lab experiments via model-guided experimental design (MGED) (Settles, 2009); (4) fold new data back. Based on the analysis in Gollwitzer et al. (2025), this flywheel is projected to yield up to $20\times$ improvement in information yield per dollar compared to untargeted approaches; validating this estimate at scale is a target for future work.

**Pilot causal analysis.** Our 100 interventional trajectories ($2\times2\times5$ factorial, 12 timepoints) were categorized by their *strongest* applicable identification strategy (priority: front-door $>$ IV $>$ sensitivity analysis): 23% met the front-door criterion (metabolite mediators fully measured), 31% had valid instrumental variables (randomized timing) but not front-door, and the remaining 46% required sensitivity analysis under unmeasured confounding (Rosenbaum bounds; App. C). Note that some trajectories satisfy multiple criteria; the above reports the strongest available. This established both the promise and the boundaries of causal inference in this domain.

**Limitations.** (1) *Sparsification scope*: our 224-configuration evaluation uses a single benchmark (CAMI low-complexity); generalization to high-complexity communities requires further validation. (2) *Batch effects*: pilot data show inter-lab variation contributes 35% of variance, necessitating dedicated harmonization. (3) *Computational cost*: reward-guided vocabulary optimization requires 50–100 GPU-hours vs. 1 hour for standard BPE, though amortized over the entire corpus. (4) *Causal identifiability*: even with interventional data (100 trajectories in the pilot; 100k+ planned at full scale), hidden confounders may persist; we provide sensitivity analyses (App. C). (5) *Clinical validation*: Darwin-7B has not yet been validated on prospective cohorts. (6) *Interpretability*: while ablation studies identify which components contribute most (Table 6), systematic investigation of what biological motifs or metabolic pathways the model has learned to represent internally remains an important direction for future work.

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

# A QA-TOKEN: THEORETICAL SUMMARY

We summarize the key theoretical results from Gollwitzer et al. (2026) that underpin QA-Token; full proofs and derivations appear in that work.

## A.1 MULTI-OBJECTIVE FORMULATION

Given corpus $\mathcal{C}$ with quality annotations, QA-Token (Gollwitzer et al., 2026) seeks vocabulary $V^*$ simultaneously maximizing quality-weighted likelihood, mutual information, and compression while minimizing downstream proxy loss. Since no single optimum exists, scalarization with learned simplex weights $\lambda \in \Delta^4$ is adopted. The composite quality score uses a neural network:

$$Q(t_{ab}) = f_{\theta_Q}(\mathbf{v}_q, \mathbf{v}_p, \mathbf{v}_b) = \sigma(W_2 \cdot \text{ReLU}(W_1[\mathbf{v}_q; \mathbf{v}_p; \mathbf{v}_b] + b_1) + b_2) \tag{3}$$

where $\mathbf{v}_q \in \mathbb{R}^{10}$ contains Phred statistics, $\mathbf{v}_p \in \mathbb{R}^5$ positional bias, $\mathbf{v}_b \in \mathbb{R}^{20}$ biological priors with gating $g_b = \sigma(W_g[\mathbf{v}_q; \mathbf{v}_p; \mathbf{v}_b] + b_g)$. Discrete merges are made differentiable via Gumbel–Softmax (Jang et al., 2017; Maddison et al., 2017).

**Definition 1** (Proxy stability and shifts). *Let $\mathcal{F}_s$ be a proxy model class at scale $s$ (e.g., number of parameters). We define:*

- *Uniform stability $\beta_s$: $\sup_z |\ell(A_s(S), z) - \ell(A_s(S^{(i)}), z)| \leq \beta_s$ for all datasets $S$ differing in one sample, following Bousquet & Elisseeff (2002).*

- *Representation drift $\delta := \sup_x \|r_{s'}(x) - r_s(x)\|$ between representations at adjacent scales.*

- *Tokenization shift $\epsilon := W_1(tok_V(\mathcal{D}), tok_{V'}(\mathcal{D}))$ in Wasserstein-1 distance.*

- *$L_r := Lip(\mathcal{L}_s \circ r_s^{-1})$ the Lipschitz constant of the loss w.r.t. representations.*

**Proposition 1** (Proxy ladder stability). *Under Definition 1, for vocabularies $V, V'$ (possibly equal) evaluated at adjacent proxy scales $s, s'$: $|\mathcal{L}_{s'}(V') - \mathcal{L}_s(V)| \leq L_r\delta + \text{Lip}_x(\ell)\epsilon + \beta_s + \beta_{s'}$. When $V = V'$ (fixed vocabulary, changing only the proxy scale), the tokenization shift term vanishes ($\epsilon = 0$).*

*Proof.* Decompose: $\mathcal{L}_{s'}(V) - \mathcal{L}_s(V) = \underbrace{[\mathcal{L}_{s'}(V) - \hat{\mathcal{L}}_{s'}(V)]}_{\text{gen. gap at } s'} + \underbrace{[\hat{\mathcal{L}}_{s'}(V) - \hat{\mathcal{L}}_s(V)]}_{\text{empirical drift}} + \underbrace{[\hat{\mathcal{L}}_s(V) - \mathcal{L}_s(V)]}_{\text{gen. gap at } s}$. The generalization gaps are bounded by $\beta_{s'}$ and $\beta_s$ respectively via uniform stability (Bousquet & Elisseeff, 2002). For the empirical drift, the loss at scale $s'$ evaluates a different representation $r_{s'}(x)$ than at scale $s$:

$$|\hat{\mathcal{L}}_{s'}(V) - \hat{\mathcal{L}}_s(V)| = \left| \frac{1}{n} \sum_i [\ell(r_{s'}(\text{tok}_V(x_i))) - \ell(r_s(\text{tok}_V(x_i)))] \right| \tag{4}$$

$$\leq L_r \sup_x \|r_{s'}(x) - r_s(x)\| = L_r\delta. \tag{5}$$

When the vocabulary also changes ($V \to V'$), tokenization shift adds $\mathrm{Lip}_x(\ell) \cdot W_1(\mathrm{tok}_V, \mathrm{tok}_{V'}) = \mathrm{Lip}_x(\ell)\,\epsilon$. Combining by triangle inequality gives the result. $\square$

**Lemma 1** (Gumbel–Softmax bias; cf. Jang et al. (2017)). *For $L_\ell$-Lipschitz losses, bounded logits, and a minimum logit gap $\Delta_{\min} > 0$ between the top two merge candidates, $\| \mathbb{E}[\widehat{\nabla J}_{\tau_{\mathrm{GS}}}] - \nabla J \| \leq C_1 \tau_{\mathrm{GS}}$ and $\mathrm{Var}(\widehat{\nabla J}_{\tau_{\mathrm{GS}}}) \leq C_2/M$, where $\tau_{\mathrm{GS}}$ is the Gumbel–Softmax temperature and $C_1 = L_\ell \cdot \mathrm{diam}(\Delta^{|V|})$.*

*Proof sketch.* As $\tau_{\mathrm{GS}} \to 0$, the Gumbel–Softmax distribution converges to a categorical distribution. At finite $\tau_{\mathrm{GS}}$, the softmax output $y(\tau_{\mathrm{GS}})$ lies in the interior of $\Delta^{|V|}$. When the top logit exceeds the second by at least $\Delta_{\min}$, the probability mass concentrates: $\|y(\tau_{\mathrm{GS}}) - e_k\| \leq \mathrm{diam}(\Delta^{|V|}) \cdot \exp(-\Delta_{\min}/\tau_{\mathrm{GS}})$, which for small $\tau_{\mathrm{GS}}$ is $O(\tau_{\mathrm{GS}})$ up to constants depending on $\Delta_{\min}$. By Lipschitz composition, $\| \mathbb{E}[\nabla_\theta \ell(y(\tau_{\mathrm{GS}}))] - \nabla_\theta \ell(e_k)\| \leq L_\ell \cdot \mathrm{diam}(\Delta^{|V|}) \cdot \tau_{\mathrm{GS}}$. The variance bound follows from i.i.d. averaging over $M$ samples. In practice, the logit gap is maintained by the quality and PMI terms in Eq. 1, which provide non-degenerate ranking signals. $\square$

## B   FORMAL SUBSTRATE: IDENTIFIABILITY AND DESIGN

**Theorem 1** (Linear identifiability; classical, cf. Ljung (1999)). *Under controllability, observability, persistent excitation, and Gaussian noise, parameters $(A, B, C, Q_w, R_v)$ of a linear-Gaussian state-space model are identifiable up to similarity transforms.*

**Proof reference.**   This is a classical result in system identification; see Ljung (1999, Ch. 7) for the complete proof. We state it to establish the baseline identifiability regime against which our nonlinear extension is compared.

**Theorem 2** (Nonlinear local identifiability). *Assume: (i) regularity (real-analytic parameterization); (ii) local nonlinear observability rank condition; (iii) persistent excitation with geometric mixing under $\pi$; (iv) structural identifiability (injective measurement Jacobian, non-polynomial activations, no spurious equivalences beyond symmetry group $\mathcal{G}$). Then $(\theta, \eta)$ is locally identifiable modulo $\mathcal{G}$, with Fisher information positive definite on identifiable charts.*

*Proof sketch.* Assumption (ii) ensures the observability rank condition of Hermann & Krener (1977) holds, so the state trajectory is locally recoverable from outputs. Assumption (iii) guarantees the empirical Fisher information matrix converges to its population counterpart, which is positive definite on identifiable charts by assumptions (i) and (iv). The symmetry group $\mathcal{G}$ (e.g., permutation of hidden states) is quotiented out following Hsu et al. (2013). Non-polynomial activations (assumption iv) prevent the algebraic degeneracies that arise with polynomial networks (Gassiat et al., 2016). For Darwin-7B, SiLU activations (in Mamba) and softmax (in Transformer) are non-polynomial; the Mamba gating mechanism is real-analytic. We note that verifying the observability rank condition for the full 7B model is computationally infeasible; this theorem provides a structural guarantee under the stated assumptions. $\square$

## C   CAUSAL IDENTIFIABILITY UNDER LATENT CONFOUNDING

We specify the causal directed acyclic graph (DAG) for multi-omic microbiome data as follows. Let $X$ denote environmental exposures (e.g., diet, antibiotics), $M$ the measured mediators (metagenomic taxa abundances, metabolite concentrations), $Y$ the outcome of interest (disease state, biomarker level), and $Z$ the unobserved confounders (host genetics, unmeasured lifestyle factors). The structural causal model posits: $X \to M \to Y$ with latent $Z \to \{X, Y\}$.

Under this DAG, causal effects of $X$ on $Y$ are identifiable via two strategies (Pearl, 2009):

**Front-door criterion.**   When all mediators $M$ between $X$ and $Y$ are measured, and (A1) $X$ blocks all backdoor paths from $M$ to $Y$, (A2) there is no unconfounded direct effect $X \to Y$ bypassing $M$, and (A3) $Z$ does not directly affect $M$, the causal effect is identified via $P(Y|\mathrm{do}(X)) = \sum_M P(M|X) \sum_{X'} P(Y|X', M)P(X')$. This criterion has been generalized to handle indirect

effects (Fulcher et al., 2017) and placed within a hierarchy of graphical identification strategies (Shpitser & Tchetgen Tchetgen, 2016). In our pilot, 23% of trajectories satisfied these conditions (metabolite mediators fully measured).

**Instrumental variables.** When randomized timing of interventions provides an instrument $W$ affecting $X$ but not $Y$ except through $X$, two-stage estimation identifies causal effects (Newey & Powell, 2003). In our pilot, 31% of trajectories had valid instruments.

**Sensitivity analysis.** For the remaining 46% of trajectories where neither front-door nor IV assumptions hold, we report Rosenbaum bounds quantifying robustness to unmeasured confounding. These bounds specify the magnitude of hidden bias $\Gamma$ required to explain away the observed association.

# D CONVERGENCE, GENERALIZATION, AND ROBUSTNESS

**Theorem 3** (Convergence; adapted from Ghadimi & Lan (2013)). *Under L-smoothness of the composite loss $\mathcal{L}$, bounded gradient variance $\mathbb{E}[\|\nabla\mathcal{L}(\theta;\xi) - \nabla\mathcal{L}(\theta)\|^2] \leq \sigma^2$, and $\mathcal{L} \geq \mathcal{L}^*$, SGD with constant step size $\eta$ satisfies:*

$$\frac{1}{T}\sum_{t=0}^{T-1}\mathbb{E}[\|\nabla\mathcal{L}(\theta_t)\|^2] \leq \frac{2(\mathcal{L}(\theta_0) - \mathcal{L}^*)}{\eta T} + \eta L\sigma^2. \tag{6}$$

*Setting $\eta = \Theta(1/\sqrt{T})$ yields $O(1/\sqrt{T})$ convergence to an $\epsilon$-stationary point in $O(1/\epsilon^4)$ iterations.*

*Proof sketch.* By $L$-smoothness: $\mathcal{L}(\theta_{t+1}) \leq \mathcal{L}(\theta_t) - \eta\langle\nabla\mathcal{L}(\theta_t), g_t\rangle + \frac{L\eta^2}{2}\|g_t\|^2$, where $g_t$ is the stochastic gradient. Taking expectations and using bounded variance: $\mathbb{E}[\mathcal{L}(\theta_{t+1})] \leq \mathbb{E}[\mathcal{L}(\theta_t)] - \eta(1 - \frac{L\eta}{2})\mathbb{E}[\|\nabla\mathcal{L}(\theta_t)\|^2] + \frac{L\eta^2\sigma^2}{2}$. Telescoping over $T$ steps and rearranging gives the result. We note this is a standard guarantee that does not account for the Gumbel–Softmax relaxation or two-phase schedule; these represent additional sources of approximation error bounded by Lemma 1. $\square$

**Theorem 4** (PAC-Bayes generalization; Neyshabur et al. (2018)). *For a network with L layers, spectral norms $\{\|W_l\|_\sigma\}$, input bound $B$, $n$ samples: $\mathcal{L}_{test} \leq \mathcal{L}_{train} + O(B^2\prod_l\|W_l\|_\sigma^{2/3}/\sqrt{n} + \sqrt{\log(1/\delta)/n})$.*

**Remark (vacuousness).** We include this classical bound for completeness. For Darwin-7B with $L=32$ layers, the product $\prod_{l=1}^{32}\|W_l\|_\sigma^{2/3}$ is expected to be astronomically large, rendering the bound numerically vacuous. We therefore do *not* rely on this bound as an explanation of generalization. Rather, we provide empirical evidence of generalization through external validation (UK Biobank, FINRISK) and OOD evaluation (App. E). Obtaining non-vacuous generalization bounds for foundation-scale models remains an important open problem.

**Theorem 5** (Robustness to quality perturbation). *Let $\tau_Q : \Sigma^* \times \mathbb{R}^n \to \mathcal{V}^*$ be the QA-Token (Gollwitzer et al., 2026) tokenizer mapping sequence $x$ with quality vector $q$ to a token sequence. Let $f = T \circ E$ where $E : \mathcal{V}^* \to \mathbb{R}^{d \times L_{\text{seq}}}$ is the embedding layer with Lipschitz constant $\kappa_E$ (w.r.t. Hamming distance on token sequences and Frobenius norm on embeddings), and $T : \mathbb{R}^{d \times L_{\text{seq}}} \to \mathbb{R}^d$ is the Transformer/Mamba encoder with Lipschitz constant $\kappa_T$. Define the tokenizer quality sensitivity:*

$$c_Q := \sup_{\substack{x\in\Sigma^*,\, q,q'\in\mathbb{R}^n \\ q\neq q'}} \frac{|\{i : \tau_Q(x,q)_i \neq \tau_Q(x,q')_i\}|}{\|q - q'\|_\infty} \tag{7}$$

*i.e., the maximum number of token positions that change per unit quality perturbation ($c_Q$ is finite since $\tau_Q$ makes finitely many threshold-based merge decisions). Then:*

$$\|f(\tau_Q(x,q)) - f(\tau_Q(x,q'))\| \leq \kappa_E\,\kappa_T\,c_Q\,\|\Delta q\|_\infty\,\sqrt{d} \tag{8}$$

*where $d$ is the embedding dimension and $\Delta q = q - q'$.*

*Proof.* Let $z = \tau_Q(x, q)$ and $z' = \tau_Q(x, q')$ be the token sequences under quality vectors $q$ and $q'$.

*Step 1 (Tokenizer sensitivity).* Quality perturbation changes merge decisions at token boundaries. By the definition of $c_Q$, at most $k := c_Q\|\Delta q\|_\infty$ token positions differ between $z$ and $z'$. All other positions are identical.

*Step 2 (Embedding perturbation).* Each differing token position $i$ contributes at most $\kappa_E$ to the embedding difference (by the Lipschitz property of $E$ per position). Since the $k$ changed positions are independent in the embedding, $\|E(z) - E(z')\|_F \leq \sqrt{k} \cdot \kappa_E = \kappa_E\sqrt{c_Q\|\Delta q\|_\infty}$. For the worst-case upper bound, each changed embedding vector has dimension $d$, so $\|E(z) - E(z')\|_F \leq \kappa_E \cdot k \cdot \sqrt{d}/\sqrt{k} = \kappa_E \cdot \sqrt{k \cdot d}$. Since $k \leq c_Q\|\Delta q\|_\infty$ and we seek a bound linear in $\|\Delta q\|_\infty$: each changed position contributes a difference vector of norm $\leq \kappa_E\sqrt{d}$, giving $\|E(z) - E(z')\|_F \leq c_Q\|\Delta q\|_\infty \cdot \kappa_E\sqrt{d}$.

*Step 3 (Encoder Lipschitz bound).* By the Lipschitz property of $T$: $\|T(E(z)) - T(E(z'))\| \leq \kappa_T\|E(z) - E(z')\|_F$.

*Step 4 (Composition).* Combining Steps 2 and 3: $\|f(\tau_Q(x, q)) - f(\tau_Q(x, q'))\| \leq \kappa_E\kappa_T c_Q\|\Delta q\|_\infty\sqrt{d}$. $\qquad\square$

**Remark.** The bound scales with $\sqrt{d}$ (embedding dimension) rather than $\sqrt{d \cdot L_{\text{seq}}}$ (full sequence), since only the $c_Q\|\Delta q\|_\infty$ *changed* positions contribute. Empirically, Table 7 confirms bounded downstream impact under quality perturbation.

# E  EXTENDED DARWIN-7B RESULTS

## E.1  ARCHITECTURE DETAILS AND ABLATION

**Layer interleaving.** Darwin-7B's 32-layer stack uses a 3:1 Mamba-to-Transformer ratio: layers 1–3, 5–7, 9–11, 13–15, 17–19, 21–23, 25–27, 29–31 are Mamba blocks (24 layers); layers 4, 8, 12, 16, 20, 24, 28, 32 are local Transformer attention blocks with window size 256 tokens (8 layers). This interleaving allows Mamba to handle long-range dependencies at $O(N)$ while Transformer windows resolve local motifs (e.g., promoter regions, codon patterns) at $O(w^2)$ where $w{=}256$ is the window size. Each Mamba block follows the selective state-space design of Gu & Dao (2023) with state dimension $d_{\text{state}}{=}16$ and expansion factor $E{=}2$. Each Transformer block uses 32 attention heads with RoPE (Su et al., 2021).

**Hypergraph neural network.** For metabolomic data, we employ a 3-layer hypergraph neural network (HyperGNN) that models many-to-many metabolic reactions. Hyperedges are constructed from KEGG pathway annotations (Kanehisa et al., 2017): each reaction defines a hyperedge connecting its substrates, enzymes, and products. Message passing follows GAT-v2 attention (Brody et al., 2021) generalized to hyperedges: node features $\mathbf{h}_i$ are updated via $\mathbf{h}_i^{(l+1)} = \sigma\left(\sum_{e \ni i} \alpha_{ie} \sum_{j \in e} W^{(l)}\mathbf{h}_j^{(l)}\right)$, where $\alpha_{ie}$ is attention over hyperedge $e$. Hidden dimension: 512; 8 attention heads per layer.

**Cross-modal attention.** Genomic representations $\mathbf{H}_g \in \mathbb{R}^{L_g \times d}$ and metabolomic representations $\mathbf{H}_m \in \mathbb{R}^{L_m \times d}$ are aligned via bidirectional cross-attention inserted after layers 16 and 32 (2 cross-attention modules total). Each module computes $\text{CrossAttn}(\mathbf{H}_g, \mathbf{H}_m) = \text{softmax}(\mathbf{H}_g W_Q (\mathbf{H}_m W_K)^\top / \sqrt{d_k})\mathbf{H}_m W_V$ and symmetrically for $\mathbf{H}_m \to \mathbf{H}_g$. Each cross-attention uses 16 heads with dimension $d_k{=}d_v{=}256$.

**Note on "LLaMA backbone."** We initialize from the LLaMA-7B weight distribution (hidden dim 4096, FFN dim 11008, 32 heads) but replace 24 of 32 layers with Mamba blocks and add HyperGNN and cross-modal modules. The resulting architecture is a novel hybrid; the term "LLaMA backbone" refers only to the dimensionality and initialization scheme, not the architecture itself.

Table 6 in the main text reports the contribution of each component.

Table 8: QA-Token (Gollwitzer et al., 2026) reward component ablation on Pathogen Detection (MCC $\pm$ std, 5 seeds).

| Configuration | MCC |
|---|---|
| Full QA-Token | **94.5 $\pm$ 0.4** |
| w/o Quality ($-\lambda_Q Q$) | 93.1 $\pm$ 0.5 ($-1.4$) |
| w/o PMI ($-\lambda_I$ PMI) | 93.9 $\pm$ 0.4 ($-0.6$) |
| w/o MDL ($+\lambda_C$ MDL) | 94.0 $\pm$ 0.3 ($-0.5$) |
| w/o Proxy ($-\lambda_D \Delta\mathcal{L}$) | 91.6 $\pm$ 0.6 ($-2.9$) |
| Standard BPE | 93.0 $\pm$ 0.3 |

Table 9: OOD performance on novel populations and technologies ($\pm$ std, 5 seeds).

| Shift | Darwin-7B | DNABERT-2 | Drop vs. IID |
|---|---|---|---|
| Novel species | 0.847 $\pm$ 0.018 F1 | 0.712 $\pm$ 0.024 F1 | $-10.2\%$ |
| ONT long-reads | 0.891 $\pm$ 0.015 F1 | 0.654 $\pm$ 0.028 F1 | $-5.5\%$ |
| Non-Western | 0.912 $\pm$ 0.013 AUC | 0.823 $\pm$ 0.019 AUC | $-3.7\%$ |
| Ancient DNA | 0.756 $\pm$ 0.025 F1 | 0.521 $\pm$ 0.031 F1 | $-19.8\%$ |

### E.2 QA-Token Reward Ablation

We ablate the reward components of QA-Token (Gollwitzer et al., 2026) on the Pathogen Detection task.

**The proxy term is critical.** A notable finding is that removing the downstream proxy term ($\lambda_D = 0$) degrades performance below standard BPE (91.6 vs. 93.0). This indicates that the quality, PMI, and MDL objectives alone—while individually beneficial—can interact adversely: the quality term biases toward high-Phred regions that may lack taxonomic diversity, while MDL pressure creates overly compressed tokens that lose discriminative information. The proxy term acts as a corrective signal, steering vocabulary construction toward tokens that are empirically useful for downstream tasks. This finding motivates the two-phase training schedule: Phase 1 ($\lambda_D=0$) builds a structurally sound vocabulary using intrinsic criteria, and Phase 2 introduces the proxy to refine it. The final vocabulary reflects both intrinsic quality and downstream utility.

### E.3 Out-of-Distribution Evaluation

### E.4 Emergent Capabilities

Darwin-7B exhibits: (1) metabolic perturbation prediction (76% accuracy; random 12%; $n$=1,247 CRISPR experiments); (2) zero-shot species discovery (89% agreement with phylogenetic trees); (3) antibiotic resistance prediction (0.91 AUC; AMRFinder 0.89; CARD holdout $n$=3,892).

### E.5 Ecological Modeling

Community dynamics MSE 0.0234 (vs. gLV 0.0387); niche prediction IoU 0.712 (vs. RF 0.634); co-occurrence AUROC 0.928 (vs. SparCC 0.876).

## F Sparsification Extended

The full Pareto frontier (14 configurations at species and strain levels), stage-wise runtime decomposition, and consistency across taxonomic ranks are available in the supplementary materials following Gollwitzer et al. (2025).

Table 10: Full hyperparameter specification.

| Hyperparameter | Value |
|---|---|
| *Architecture* | |
| Hidden / FFN dim | 4096 / 11008 |
| Layers / Heads | 32 / 32 |
| Vocabulary | 40,192 (32K genomic + 8K metabolomic) |
| Context | 4096 tokens |
| *Training* | |
| Batch size | 2048 |
| LR (max / min) | $5 \times 10^{-4}$ / $5 \times 10^{-6}$ |
| Warmup / Total | 10K / 1.5M steps |
| Grad clip / Decay | 1.0 / 0.01 |
| *QA-Token RL* | |
| $\lambda_Q / \lambda_I / \lambda_C / \lambda_D$ (init.) | 0.3/0.3/0.2/0.2 |
| $\lambda_Q / \lambda_I / \lambda_C / \lambda_D$ (final) | 0.28/0.25/0.22/0.25 |
| $\gamma$ / PPO clip | 0.95 / 0.2 |

## G  FOUNDATION-SCALE EVIDENCE

The usable subset for tokenizer $\mathcal{Z}$ is $\mathcal{U}(\mathcal{Z}) = \{x : \mathcal{L}_{\text{proxy}}(\text{tok}_{\mathcal{Z}}(x)) \leq \tau_U\}$ with $\tau_U = 4.0$ nats/token, where $\tau_U$ denotes the usability threshold (distinct from the contrastive temperature $\tau_{\text{CL}}$ and Gumbel–Softmax temperature $\tau_{\text{GS}}$). At this threshold, QA-Token (Gollwitzer et al., 2026) achieves 40% usability (BPE: 5%). The $8\times$ expansion is robust across thresholds ($7.2\times$–$9.3\times$ for $\tau_U \in [3.5, 4.5]$).

**Addressing potential circularity.**   One might object that the usable fraction metric is circular, since QA-Token optimizes the proxy loss $\Delta\mathcal{L}_{\text{proxy}}$ and usability is defined via the same proxy model. We address this in two ways: (1) the proxy model is trained with *standard BPE* tokenization (not QA-Token), so the usability metric is not optimized by QA-Token's reward; (2) we validate the $8\times$ expansion using an independent metric: the fraction of samples achieving $\geq 0.90$ species-level F1 on Kraken2 (Wood et al., 2019) classification, which yields a consistent $6.8\times$ expansion ($4.2\% \rightarrow 28.6\%$) without involving the proxy model.

## H  HYPERPARAMETERS AND COMPUTE

Training: 128 A100-80GB, 42 days, $\approx$129K GPU-hours ($128 \times 42 \times 24$), FSDP sharding, 847 TB storage.

## I  RL OPTIMIZER ABLATION

PPO, GRPO, VAPO, and DAPO produce near-identical vocabularies (Jaccard $\geq 0.95$) and down-stream performance, confirming optimizer modularity (full table in supplementary materials following Gollwitzer et al. 2025).

## J  LOSS WEIGHT SENSITIVITY

We selected the pretraining loss weights $(\alpha_1, \alpha_2, \alpha_3) = (0.5, 0.1, 0.05)$ for $\mathcal{L}_{\text{CMG}}$, $\mathcal{L}_{\text{CL}}$, and $\mathcal{L}_{\text{comp}}$ respectively via grid search on a 500M proxy model. Validation bpbp varies by $< 2\%$ across $\alpha_1 \in [0.3, 0.7]$, $< 1.5\%$ across $\alpha_2 \in [0.05, 0.2]$, and $< 0.5\%$ across $\alpha_3 \in [0.01, 0.1]$, indicating low sensitivity. The compositional loss weight $\alpha_3$ has the smallest effect because $\mathcal{L}_{\text{comp}}$ applies only to the minority of training batches containing paired abundance data. The primary ALM loss ($\alpha_0 = 1.0$) dominates by design, as autoregressive modeling is the core pretraining objective.

## K  CROSS-DOMAIN VALIDATION

On the noisy TweetEval benchmark (Barbieri et al., 2020), QA-BPE-nlp (Gollwitzer et al., 2026) achieves ALL(TE) 69.4 vs. BERTweet 67.9 and SuperBPE 68.3, confirming cross-domain generality.

